# 1 × 4 Wavelength Demultiplexer C-Band Using Cascaded Multimode Interference on SiN Buried Waveguide Structure

**DOI:** 10.3390/ma15145067

**Published:** 2022-07-21

**Authors:** Jonathan Menahem, Dror Malka

**Affiliations:** Faculty of Engineering, Holon Institute of Technology (HIT), Holon 5810201, Israel; jonathanmenahem@gmail.com

**Keywords:** BPM, FDTD, MMI, WDM, buried waveguide, SiN, PIC

## Abstract

Back reflection losses are a key problem that limits the performances of optical communication systems that work on wavelength division multiplexing (WDM) technology based on silicon (Si) Multimode Interference (MMI) waveguides. In order to overcome this problem, we propose a novel design for a 1 × 4 optical demultiplexer based on the MMI in silicon nitride (SiN) buried waveguide structure that operates at the C-band spectrum. The simulation results show that the proposed device can transmit four channels with a 10 nm spacing between them that work in the C-band with a low power loss range of 1.98–2.35 dB, large bandwidth of 7.68–8.08 nm, and good crosstalk of 20.9–23.6 dB. Thanks to the low refractive index of SiN, a very low back reflection of 40.57 dB is obtained without using a special angled MMI design, which is usually required, using Si MMI technology. Thus, this SiN demultiplexer MMI technology can be used in WDM technique for obtaining a high data bitrate alongside a low back reflection in optical communication systems.

## 1. Introduction

The vast growth of developments in the optical communication systems over the C-band spectrum requires new and powerful waveguide components that can support high-speed light communication [1] with low power losses, large bandwidth, and low back reflection losses [2,3].

Optical demultiplexers are an important part of communication networks and can be implemented with different technology techniques: Y-branch devices [4,5,6], Mach–Zehnder interferometers [7], and Multimode Interference (MMI) couplers [8,9,10]. Wavelength division multiplexing (WDM) technology is used to increase the data transfer bitrate by decreasing the spacing between peak wavelengths, and as a result, more channels can be utilized for a single spectral band [11].

A buried waveguide is a simple structure built from a narrow high index region surrounded by a cover of low index material [12]. This structure allows light to be strongly confined and guided through it due to the total internal reflection effect [13].

MMI coupler devices are commonly used in photonic integrated circuits (PICs) due to their large optical bandwidth, low losses [14], and compact structure [15,16]. MMI waveguide operation is based on the self-imaging effect, by which the electric field profile that enters the device is duplicated into multiple images at periodic intervals along the propagation axis of the device [17]. 

One of the key problems that can affect the quality of the transmitter system is the back reflection, especially the reflection of light in the opposite direction into the laser source. The silicon (Si) MMI couplers can suffer from reflections because of the self-imaging effect and also from the mismatch between the refractive indexes of Si and silica (SiO_2_) [18]. 

Research studies show the use of polycarbonate polymer optical fiber as an RGB multiplexer/demultiplexer with transmission losses of 0.6 to 1.2 dB [19,20]. In addition, a gallium nitride (GaN) multi-slot waveguide structure was used to function as a four-green light demultiplexer in the visible range [21]. MMI based on GaN material was used as a demultiplexer for dividing four channels in the visible light spectrum [22] and eight channels in the C-band spectrum [23]. However, in these studies, the back reflection effect was not addressed, despite its being a critical parameter for the transmitter performances 

In general, the MMI coupler can suffer from various kinds of reflections that can be grouped into two types. The first type of reflection is internal resonance modes, which are caused by several self-imaging co-occurs. The second type of reflection is a reflection back into the access waveguides where there is a phase difference in the input to the MMI that can lead to imaging of the input back to itself.

To solve this issue, a lower refractive index waveguide material that supports the C-band range with low absorption was used, and the material is silicon nitride (SiN). In addition, the study of the light coupling mechanism of SiN MMI couple waveguides was demonstrated, for the first time, that it can be used as a four-channel demultiplexer device. Moreover, researchers show the great potential of using SiN-buried-waveguide-based MMI coupler structures [24] for designing splitters and wavelength demultiplexers in the C-band range [13] due to their low back reflection loss and low thermal sensitivity, meaning that there is a very slight change in the operating wavelength for every degree Celsius [25].

Previous research has shown the possibility of obtaining a very low insertion loss using SiN by designing a device based on Mach–Zehnder interferometer lattice wavelength demultiplexer [26] but with relatively high crosstalk. Moreover, the back reflection losses were not discussed.

In this paper, we present a design of a 1 × 4 wavelength demultiplexer based on an MMI coupler in a SiN buried waveguide structure that divides four channels in the C-band light range. The selected wavelengths were found to be 1530, 1540, 1550, and 1560 nm. 

The device design is based on a cascade of three 1 × 2 MMI couplers, three input waveguide segments and tapers, six S-bends, and six output tapers. The geometrical dimensions of the buried waveguide structure and the MMI couplers were analyzed to obtain the self-imaging effect and to find the optimal parameters of the MMI couplers. The simulations were carried out by using the beam propagation method (BPM) [27,28,29] combined with finite difference time domain (FDTD), which was analyzed and processed by using Python scripts. 

## 2. The 1 × 4 Demultiplexer Design and Theoretical Aspect

Figure 1a demonstrates the XY plane cross-sectional view at Z = 0, where the red color area represents SiN, and the magenta color area represents the SiO_2_ cover. H_strip_ is the height of the SiN layer, and W_srtip_ represents its width. Moreover, n_strip_ and n_cover_ are the refractive indexes of the SiN strip, and the SiO_2_ cover, and their values are 1.989 and 1.444, respectively.

Figure 1b demonstrates the XZ plane cross-sectional view at Y = 0, where the three cascaded MMI couplers designed for the 1 × 4 demultiplexer device are presented. The width of the MMI coupler is W_MMI_, and their lengths are L_MMI1_, L_MMI2_, and L_MMI3_. 

The width of the input waveguide segment was chosen to be 500 nm, with a height of 310 nm. The length of the input tapers for MMI1, MMI2, and MM3 couplers was set to be 40 μm, and the width varied between 0.5 and 0.75 μm. The length of the output tapers for MM1, MMI2, and MMI3 couplers was set to be 35 μm, and the width varied between 0.9 and 0.5 μm. The gap (G_t_) between the output tapers for each MMI coupler is 0.78 μm.

The MMI1 coupler in the three cascade is designed to be suitable for dividing four wavelengths, which are 1530 nm (λ_1_), 1540 nm (λ_2_), 1550 nm (λ_3_), and 1560 nm (λ_4_). The pair (λ_2_/λ_4_) is propagated into the MMI2 coupler, and the pair (λ_1_/λ_3_) is propagated into the MMI3 coupler, as shown in Figure 1b.

The width of S-bends was chosen to match the output width of the taper, which was set to 0.5 μm, with a length that varies from 69 to 80 μm. The distance between the outputs of the S-bends (G_s_) was chosen to be 16.5 μm for MMI1 and 11.5 μm for MMI2 and MMI3. The full device length is 6.63 mm.

According to the self-imaging effect, every wavelength that enters the multimode region of the device produces a direct or mirrored image of itself periodically. The distance from the entry to the point of the first image is called the beat length ($L_{\pi }$), and it is given by the following [14]:(1)$$L_{\pi }\;\approx \;\frac{{4n}_{\mathrm{eff}}W_{\mathrm{eff}}^{2}}{{3\lambda }_{n}}\;;\;n=1,2,3,4$$
where λ_n_ stands for the operating wavelengths for n = 1,2,3,4, and n_eff_ is the effective refractive index of SiN for the electrical fundamental mode. This parameter was calculated by using the BPM mode solver. The effective width of the MMI coupler is W_eff_, which takes into account the lateral penetration depth of each mode field at the strip boundaries. In the case of transverse electric (TE) mode, the W_eff_ size is given by the following [14]:(2)Weff=WMMI+(λnπ)(neff2−ncover2)−12
where W_MMI_ is the physical width of the MMI coupler, as seen in Figure 1b.

In order to separate two pairs of different wavelengths by using the MMI1, MMI2, and MMI3 couplers, these conditions must be met for each coupler: (3)$$L_{\mathrm{MMI}2}{=p}_{1}L_{\pi }^{\lambda_{1}}{=(p}_{1}{+q}_{1}{)\;L}_{\pi }^{\lambda_{3}}{\;\;;\;\;L}_{\mathrm{MMI}3}{=p}_{2}L_{\pi }^{\lambda_{2}}{=(p}_{2}{+q}_{2}{)\;L}_{\pi }^{\lambda_{4}}$$
(4)$$L_{\mathrm{MMI}1}={\;p}_{3}L_{\pi }^{\lambda_{1}}=\left(p_{3}+q_{3}\right){\;L}_{\pi }^{\lambda_{2}}=\left(p_{3}+q_{3}+1\right){\;L}_{\pi }^{\lambda_{3}}=\left(p_{3}+q_{3}+2\right){\;L}_{\pi }^{\lambda_{4}}$$
where q is an odd number, and p is an integer. 

It is possible to shorten the propagation distance by a factor of three by canceling the third mode from inside the MMI. In order to do so, the input taper of the MMI must be shifted by an offset of ± (1/6) W_eff_ from its center. Moreover, these conditions can be optimized to obtain better performances by using BPM simulations. 

The insertion loss of the demultiplexer is given by the following:(5)$$\mathrm{Loss}(\mathrm{dB})=–10\log\left(\frac{P_{\mathrm{out}}}{P_{\mathrm{in}}}\right)$$
where P_in_ is the input power, and P_out_ is the output power.

The crosstalk loss of the demultiplexer is given by the following:(6)$${\mathrm{CT}}_{n}=\frac{1}{3}\overset{4}{\underset{m=1}{\sum }}10\log\left(\frac{P_{m}}{P_{n}}\right)$$
where P_n_ is the desirable port power, and P_m_ is the power that is interfering in the other ports. 

To minimize the bend loss, the dimensions of the S-bend regions were chosen carefully. According to Zamhari and Ehsan, the optimal S-bend offset is 5 μm for Si [30]. Therefore, in our case, the offset is around 5 μm, and the radius of the S-bend is given by the following equation:(7)$$R=\frac{1}{O}(\frac{L^{2}{+O}^{2}}{4})$$
where O is the S-bend offset, and L is the S-bend length.

## 3. Results

The simulations of the three MMI couplers and the buried structure were performed by using Rsoft photonic CAD software based on BPM and FDTD tools, while python scripts were used to process the results data to find the optimal values. 

Figure 2a shows the TE fundamental mode profile inside the SiN strip for an operating wavelength of 1530 nm at the XY plane, and Figure 2b shows the horizontal cut (Y = 0.15 μm). The strong confinement is shown with the red color, and the spot size of the mode is 2.36 μm × 2.22 μm, which can be increased by using an adiabatic taper to obtain better coupling between the device and the laser source. A similar mode profile is obtained for all the other operating wavelengths. The mode solution was used as the launch condition for the cascaded MMI couplers at the input waveguide.

The effective refractive indexes (n_eff_) values were calculated and found by solving the fundamental mode for each of the operating wavelengths, as shown in Table 1.

Figure 3 shows the optimizations for the selected optimal value of the SiN height, which is 320 nm, and the tolerance range ±20 nm (which is ±6.25% from the optimal value) around the optimal value to obtain normalized power (relative to the input power of the whole device) 46–64%. As can be seen in Figure 3, the max power is obtained for the 310 nm height value. However, from a fabrication point of view, this value does not satisfy the limit of fabrication error that can be handled today. This limit is a result of the geometrical dimensions error for the fabrication process with a high accuracy, which is usually around ±20 nm from the optimal value. Thus, the proposed device can function well with a standard error of ±20 nm in the SiN layer thickness.

Figure 4 shows the optimal MMI coupler width value, which is 3.5 μm, and the tolerance range around the optimal value by ±20 nm (which is ±0.6% from the optimal value) to obtain normalized power (relative to the input power of the whole device) of 56–64%. This limit is chosen because of the geometrical dimensions error for the fabrication process with a high accuracy, which is usually around ±20 nm from the optimal value. These fabrication results can be utilized for the prototype test structure.

The beat length was calculated by using Equation (1) for each wavelength and using python scripts, and the parameter p was calculated to be 65 for MMI2 and MMI3, which resulted an MMI length of 2072 and 2088 μm, respectively. For MMI1, p = 131, resulting an MMI length of 4208 μm. 

Using the Equations presented above, (3) and (4), combined with BPM simulation, the optimized MMI coupler lengths were calculated. To ensure max output power with good fabrication error, L_MMI1_, L_MMI2_, and L_MMI3_ were chosen to be 4153, 2067, and 2078 μm, respectively. Figure 5a–c shows a tolerance range of 5 μm (which is ±0.12% for L_MMI1_ and ±0.24% for L_MMI2_ and L_MMI3_ from the optimal value) around the peak value for each MMI length in correlation with the relevant wavelengths to obtain normalized power (relative to the input power for each coupler independently) of 68–74%. This large tolerance range gives us good flexibility from the fabrication point of view. In other words, the MMI coupler lengths can easily deal with a large error dimension over ±250 nm without losing a significant power.

These fabrication results are shown in Figure 3, Figure 4 and Figure 5 and can be utilized for the prototype test structure for understanding the physical error of the geometrical parameter (Wmmi, Height, L_MMI1_, L_MMI2_, and L_MMI3_) to improve the experimental results.

The intensity profile of the 1 × 4 SiN MMI demultiplexer device is shown in Figure 6a–d for the operating wavelengths at the XZ plane. The light enters the input waveguide segment, where Z = 0, and travels through the input taper into MMI1 section. The division of four wavelengths into two pairs (λ_2_/λ_4_ and λ_1_/λ_3_) occurs where Z = 4243 μm (MMI1 output). Afterward, the light propagates into the next MMIs through the S-bends, and further division occurs at MMI2 output for 1540 nm (λ_2_) (b) and 1560 nm (λ_4_) (a), where Z = 6515 μm, and at MMI3 output for 1530 nm (λ_1_) (c) and 1550 nm (λ_3_) (d), where Z = 6526 μm. Finally, the light propagates into the four output ports through the S-bends, where Z = 6630 μm. These figures show the coupling length behaver of the SiN MMI coupler, which is higher compared to the Si MMI coupler, thus leading to a larger footprint size.

The spectral transmission of the four channels in the C-band (1525–1565 nm) spectrum can be seen in Figure 7. The data were calculated by solving the mode for each wavelength in this range, and then it was inserted into the optimal design for solving the demultiplexer, and the data were processed with python codes.

The crosstalk and insertion losses and full width half maximum (FWHM) were calculated by using Equations (5) and (6) combined with the normalized spectral power in the C-band for each port, as can be seen in Table 2. The crosstalk ranges between 20.908 and 23.665 dB, insertion losses range between 1.986 and 2.351 dB, and the bandwidth ranges between 7.68 and 8.08 nm. These results can be utilized in the WDM system for increasing data bitrate by four and without suffering from crosstalk. To obtain good calculation accuracy, the optimized grid size for the *x*-axis and *y*-axis was set to 50 nm, and the grid size for the *z*-axis was set to 40 nm for all BPM simulations.

Another important characteristic of the MMI coupler is the back reflection into the input segment, which can be very problematic to the laser source. In this work, we used the SiN material to minimize the back reflection power coming from the MMI coupler due to the self-imaging. 

To calculate the back reflection power, a monitor was placed in the input waveguide to collect all the light that was reflecting back from the MMI coupler, as shown in Figure 8. Back reflection losses were calculated by using FDTD simulation and are shown in Table 3. As expected, a lower back reflection is obtained through the C-band window by using the SiN MMI coupler waveguides and without the need for a special angled MMI coupler. The optimal grid size for the *x*-axis, *y*-axis, and *z*-axis was set to 10 nm to obtain good mesh convergence for all FDTD simulations.

To emphasize our SiN MMI technology advantages over other demultiplexer designs, comparison of the key characteristics between previously published works to our design was performed. Table 4 shows the comparison between the SiN MMI demultiplexer design proposed in this work and other types of demultiplexers. The key characteristics that were compared are insertion losses, crosstalk, FWHM, spectrum, and back reflection losses. As can be seen in Table 4, our design has benefits in each aspect over the other demultiplexer designs, i.e., lower insertion loss, better crosstalk, and a larger FWHM. In addition, most of the research does not include back reflection loss. Moreover, our design has a lower back reflection loss, as expected of using SiN waveguide technology that can be utilized for working with lasers that are sensitive to the back reflection effect.

## 4. Conclusions

This paper presented a design of a new and novel 1 × 4 demultiplexer in the C-band, using three cascaded MMI couplers based on a SiN buried waveguide structure. 

The results show the optimized parameters that should be used to divide four wavelengths of the MMI coupler by using the SiN buried waveguide. The wavelengths are 1530, 1540, 1550, and 1560 nm.

The total propagation length of the device is 6.63 mm, with losses ranging between 1.986 and 2.351 dB, excellent crosstalk ranging between 20.908 and 23.668 dB, and FWHM ranging between 7.68 and 8.08 nm. These results suggest that such a device could be useful in long-distance optical communication networks that use WDM technology in the C-band spectrum.

Moreover, it is shown that the proposed device has a low back reflection loss, ranging between 40 and 41 dB, without using a special angled MMI design, and this is because of the use of SiN as the core material. 

The results show the promising potential for such a device to be implemented in WDM technology communications systems to increase the data bitrate. 

Because of the good flexibility in tolerance range (±2.5 μm) for the MMI coupler lengths, the proposed device design can be fabricated by using the fab processes that are available today.

The simple design of the device makes it easier to expand the number of channels by cascading multiple MMI couplers and adjusting their geometrical parameters accordingly.

Although this device was presented as a 1 × 4 demultiplexer, the direction of light can be reversed, and it can be utilized as a 4 × 1 multiplexer.

## Figures and Tables

**Figure 1 materials-15-05067-f001:**
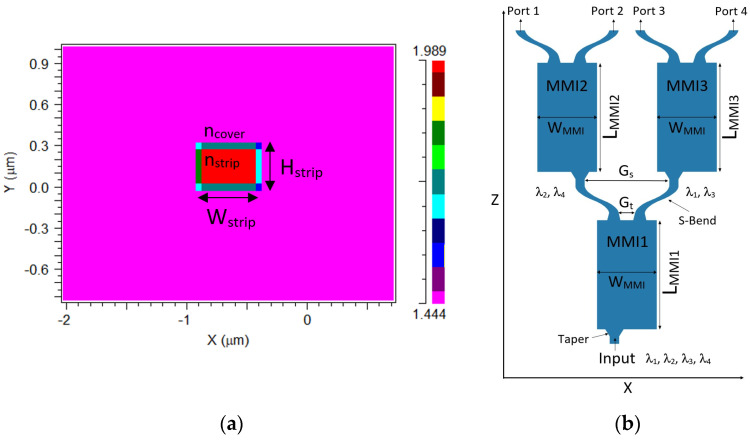
An illustration of the 1 × 4 wavelength demultiplexer: (**a**) in the XY plane alongside the refractive index bar and (**b**) in the XZ plane.

**Figure 2 materials-15-05067-f002:**
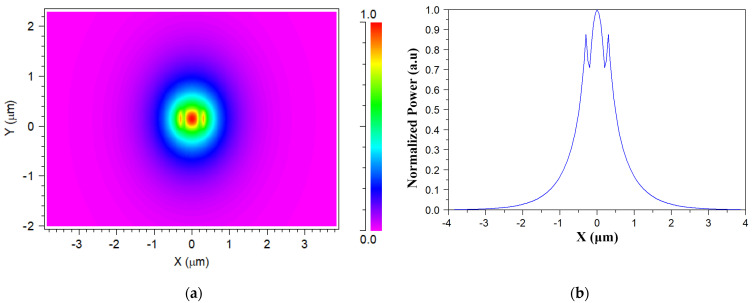
(**a**) TE fundamental mode for 1530 nm and (**b**) its horizontal cut, where Y = 0.15 μm.

**Figure 3 materials-15-05067-f003:**
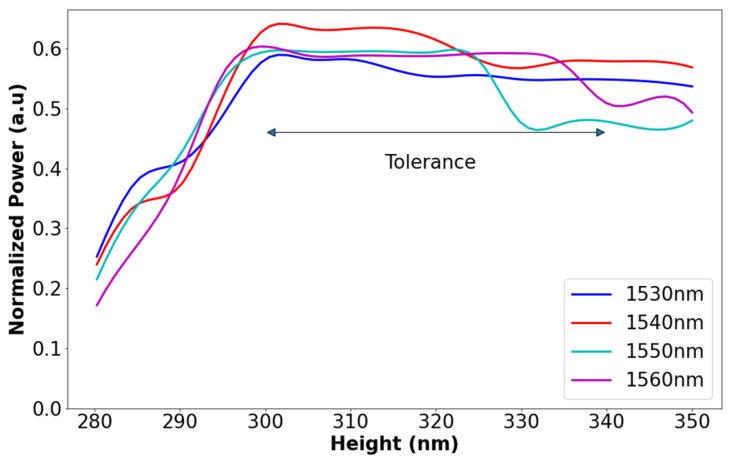
Normalized power as a function of strip height (SiN).

**Figure 4 materials-15-05067-f004:**
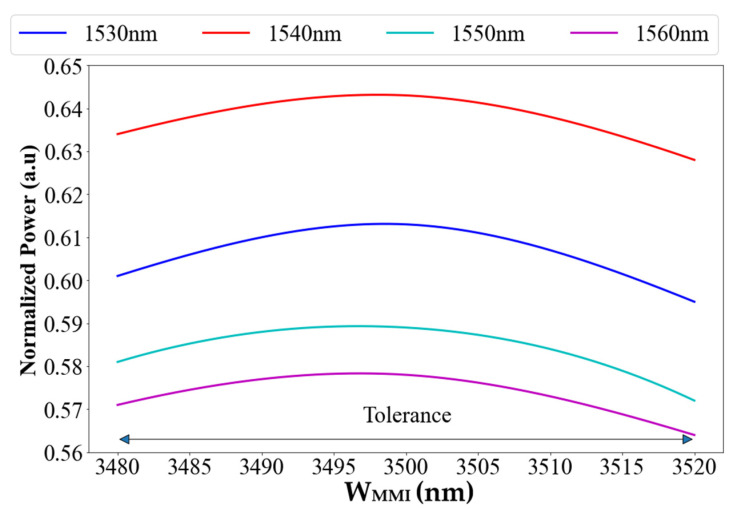
Normalized power as a function of MMI width for each wavelength.

**Figure 5 materials-15-05067-f005:**
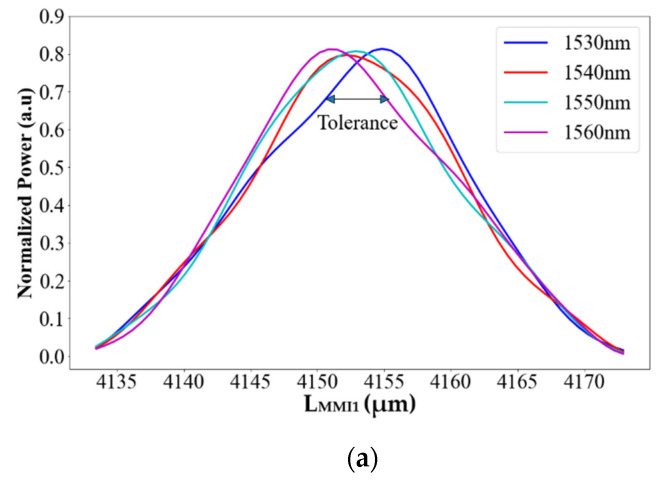
Normalized power as a function of (**a**) L_MMI1_, (**b**) L_MMI2_, and (**c**) L_MMI3_.

**Figure 6 materials-15-05067-f006:**
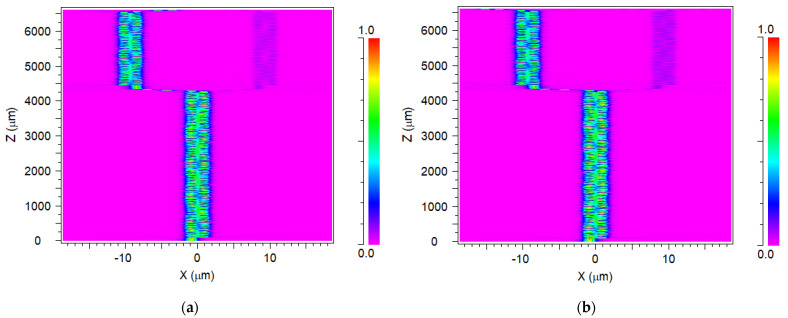
Intensity distributions of the 1 × 4 SiN MMI demultiplexer: (**a**) λ_4_ = 1560 nm (port 1), (**b**) λ_2_ = 1540 nm (port 2), (**c**) λ_1_ = 1530 nm (port 3), and (**d**) λ_3_ = 1550 nm (port 4).

**Figure 7 materials-15-05067-f007:**
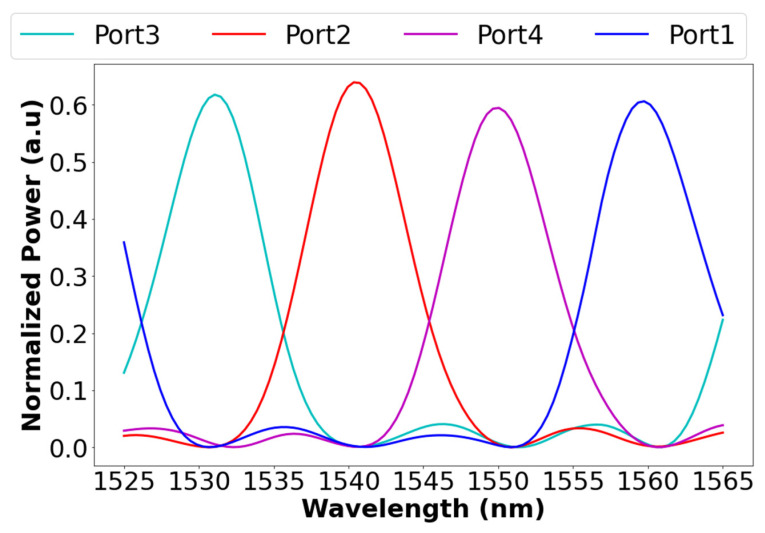
Normalized power relative to the input in each port as a function of wavelength in the C-band spectrum.

**Figure 8 materials-15-05067-f008:**
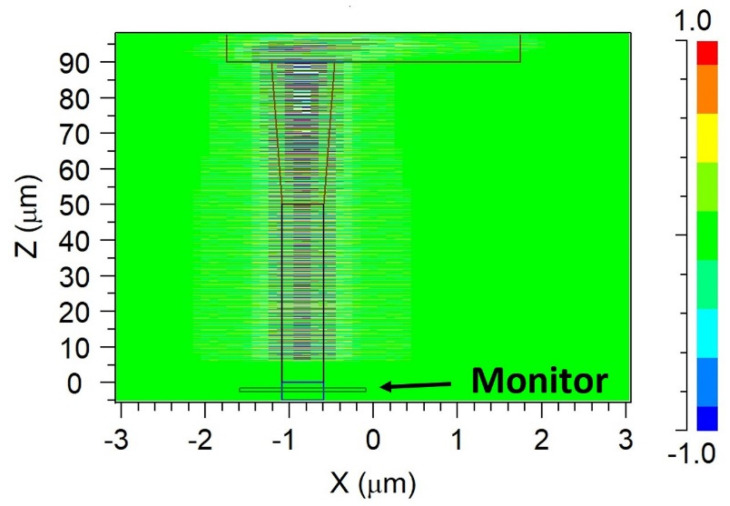
Back reflection calculation using FDTD simulations.

**Table 1 materials-15-05067-t001:** The effective refractive index values for the operating wavelengths.

λ (nm)	1530	1540	1550	1560
n_eff_	1.48388	1.48234	1.48084	1.47939

**Table 2 materials-15-05067-t002:** Crosstalk, losses, and FWHM values for each port.

λ (nm)	1530	1540	1550	1560
**Port**	3	2	4	1
**Crosstalk (dB)**	21.498	23.665	20.908	22.713
**Losses (dB)**	2.351	1.986	2.255	2.197
**FWHM (nm)**	7.68	7.68	8.08	8.08

**Table 3 materials-15-05067-t003:** Back reflection loss values for the operating wavelengths.

Wavelength (nm)	1530	1540	1550	1560
**Back Reflection (dB)**	−41	−40.8	−40.5	−40

**Table 4 materials-15-05067-t004:** Characteristics comparison between various demultiplexer designs.

Demultiplexer Type	Insertion Losses (dB)	Crosstalk (dB)	FWHM (nm)	Band Range	Back Reflection (dB)	Device Footprint (μm^2^)
1 × 4 Si modified-T demultiplexer [31]	~2.31	~21.1	~0.455	C-Band	N/A	536
1 × 8 Si MMI demultiplexer [32]	~3.09	N/A	N/A	C-Band	N/A	18 × 18,000
1 × 4 MMI GaN slot-waveguide [22]	~0.11	~22.7	~9.15	Visible Light	~36.495	3.8 × 700
1 × 4 GaN multi-slot waveguide [21]	~0.127	~24.1	~9.1	Visible Light	~36.5	3.2 × 104
1 × 4 Mach-Zehnder SiN lattice demux [13]	~2.8	~11.5	N/A	O-Band	N/A	900 × 2500
1 × 4 MMI SiN buried waveguide [in this work]	~2.197	~22.196	~7.88	C-Band	~40.57	32 × 6630

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
