# Peer review of "1 × 4 Wavelength Demultiplexer C-Band Using Cascaded Multimode Interference on SiN Buried Waveguide Structure"

_materials, 2022, doi:10.3390/ma15145067_

Round 1

Reviewer 1 Report

This manuscript proposes a design of demultiplexer in the C-band using cascaded MMI couplers based in SiN buried waveguide structure. This work is interesting to discuss and will benefit the research community. However, after reading the article carefully, we have concluded that the article has shown effort. However, it still needs some major changes before it is ready to be published.

1.           First and foremost, the topic covered by the article is very important, but it is not new, as there are many very solid scientific articles and review articles that have dealt with the same topic in the literature.

2.           The data, results and explanation presented in the article are completely insufficient to complete the content of the paper as a scientific article. Generally, result discussions can be improved. Please refrain from just reporting trends and values that can be easily read from figures and table. Instead, please provide physical interpretations and insights to justify and explain the observed trends (The explanation given to the results obtained is very superficial, it is not in-depth and detailed. The scientific explanation provided must be qualitatively improved (All the results obtained did not explain an accurate and detailed scientific explanation).

3.           The language of this work is unsatisfactorily poor. There are numerous and serious language errors spread throughout the manuscript in terms of grammar, word choice, and sentence structure. This is of paramount importance before this work can be recommended for publication in MDPI Materials.

4.           There are numerous format errors in the article. 1) Equations, 2) paragraphs, 3) subscript and so forth. Please revise the article accordingly.

5.           In the introduction section, the author compares the Silicon with the Silicon Nitride (SiN) based material for the devices.  It is suggested to also include other material that have been used as the demultiplexer waveguide device.

6.           In page 6, line 166-168, the author claimed that “This large tolerance range give us a good flexibility from the fabrication point of view.” Such statement is incomplete, where it needs to be supported by further explanation. Please justify this claim.

7.           It is recommended to also include the conventional design of Silicon -based devices to compare performance. This will strengthen the conclusions and thus highlight the novelty of the article.

Author Response

Dear Editor,

Thank you for your mail from July 4th regarding the review results of manuscript id: materials-1808104: "1 × 4 Wavelength Demultiplexer C-Band Using Cascaded Multimode Interference on SiN Buried Waveguide Structure". We have revised the manuscript while addressing all the comments made by the reviewers. We added new results and clarification text. Our detailed reply to the comments made by the reviewers can be seen below. I hope that in its revised form you may find the manuscript suitable for publication in Materials.

Reviewer 1:

This manuscript proposes a design of demultiplexer in the C-band using cascaded MMI couplers based in SiN buried waveguide structure. This work is interesting to discuss and will benefit the research community. However, after reading the article carefully, we have concluded that the article has shown effort. However, it still needs some major changes before it is ready to be published.

  1. First and foremost, the topic covered by the article is very important, but it is not new, as there are many very solid scientific articles and review articles that have dealt with the same topic in the literature.

Answer:

We thank the reviewer for his comment. Although the use of silicon nitride (SiN) material nor MMI technology is not new, the novelty is the combination of the use of SiN as the core material for the device structure and MMI technology to reduce the back reflection loss. Thus, the novelty is the study of the light coupling mechanism using SiN MMI coupler waveguides to function as a four-channel demultiplexer over the C-band. To the best of our knowledge, there are no other articles dealing with this subject.

Reviewer 1:

  1. The data, results and explanation presented in the article are completely insufficient to complete the content of the paper as a scientific article. Generally, result discussions can be improved. Please refrain from just reporting trends and values that can be easily read from figures and table. Instead, please provide physical interpretations and insights to justify and explain the observed trends (The explanation given to the results obtained is very superficial, it is not in-depth and detailed. The scientific explanation provided must be qualitatively improved (All the results obtained did not explain an accurate and detailed scientific explanation).

Answer:

 We thank the reviewer for his comment. In the revised manuscript we added a further results discussion of the scientific explanation for each figure as can be seen in the simulation results section as the reviewer suggested.

Reviewer 1:

  1. The language of this work is unsatisfactorily poor. There are numerous and serious language errors spread throughout the manuscript in terms of grammar, word choice, and sentence structure. This is of paramount importance before this work can be recommended for publication in MDPI Materials.

Answer:

We thank the reviewer for his comment and apologize for the inconvenience. In the revised manuscript, we corrected all the English grammar errors. A thorough check was performed on grammar, spelling, and punctuation for the whole manuscript.

Reviewer 1:

  1. There are numerous format errors in the article. 1) Equations, 2) paragraphs, 3) subscript and so forth. Please revise the article accordingly.

Answer:

We thank the reviewer for his comment. These issues have been treated in the revised manuscript as the reviewer's suggestion.

Reviewer 1:

  1. In the introduction section, the author compares the Silicon with the Silicon Nitride (SiN) based material for the devices. It is suggested to also include other material that have been used as the demultiplexer waveguide device.

Answer:

As the reviewer suggested, we added more materials to the introduction section such as gallium nitride and polycarbonate.

Reviewer 1:

  1. In page 6, line 166-168, the author claimed that “This large tolerance range give us a good flexibility from the fabrication point of view.” Such statement is incomplete, where it needs to be supported by further explanation. Please justify this claim.

Answer:

We thank the reviewer for his comment. We added a further explanation to the manuscript according to the reviewer's suggestion.

Reviewer 1:

  1. It is recommended to also include the conventional design of Silicon -based devices to compare performance. This will strengthen the conclusions and thus highlight the novelty of the article.

Answer:

We thank the reviewer for his comment. As the reviewer suggested, we added additional comparison to Silicon-based devices to highlight the novelty of our paper, which is low back reflection compared to silicon.

Reviewer 2 Report

The authors designed a C-band demultiplexer using a cascade of three MMIs based on computational means. They adopted a SiN waveguide buried in SiO2 to reduce the back reflection loss, and optimized the dimensions of the demultiplexer carefully to yield competitive characteristics. This work is a unique contribution to the field. However, the quality of the presentation is far below the standard of publication. For this reason, I do not recommend its publication in the journal Materials. I have the following comments for the authors to consider to improve the manuscript. 

Major comments

1. The authors wrote that their design is a novel design. Could the authors explain the novelty of their design in more explicit manner? 

2. In line 127, the authors mentioned BPM and FDTD tools of Rsoft photonic CAD software. Can the authors provide necessary simulation parameters for reproducing the results? 

3. In line 209, the authors wrote that "our design has the best back reflection loss". The description of "best" is misleading since 1) The back reflection loss of the design in this work is higher than that in Ref 28, and 2) There are only two values (not many) of back reflection loss available for comparison. 

Minor comments

4. In line 15, the verb “is” is missing between the words “which” and “usually”. 

5. The sentence in line 31 is hard to understand. The authors might want to change the verb “proposed”. 

6. In line 43, the expression “reflections can be grouped into two mechanisms” is problematic. After grouping, reflections are still reflections but not mechanisms. 

7. In line 49, “Silicon-Nitride” should be “silicon nitride” where the hyphen is symbol removed, and the initial letters are lower case. 

8. In line 50, the expression “… using a SiN [19] buried waveguide 50 based MMI coupler structures for …” is confusing. Do you mean “… using SiN-buried-waveguide-based MMI coupler structures for …”? Note that the article “a” is removed. 

9. In line 55, the initial letter of “Insertion” should be lower case. 

10. It is better to use active voice in the sentence “In this paper, … is presented.” in lines 59—61. 

11. In line 64, should “S-bands” be “S-bends”? 

12. In lines 81 and 83, the phrase “between … to …” should be either “between … and …” or “from … to …”.

13. In Fig. 1a, the colorer label is missing. 

14. In Fig. 1b, adding tics for both X and Z axes help readers to understand device dimensions. 

15. In line 96, “is producing” can be just “produces”.

16. In line 97, it is hard to understand what “the point of two lowest-order modes” is. 

17. In Eq. (1), the superscript is missing for L_\pi. 

18. In lines 99—100, the expression “the effective refractive index of the electrical fundamental mode in the SiN area” can be rewritten as “the effective refractive index of SiN for the electrical fundamental mode”. 

19. In line 105, “MMI” should be “MMI1 and MMI2”.

20. In lines 107, 115, and 117, “Where” should be “where” in which the initial letter is lower case. 

21. In lines 105—109, the two paragraphs can be a single paragraph. 

22. In line 110, one can replace “There is an option” with “It is possible”. 

23. In line 114, “losses” should be “loss”. 

24. In line 120, can the authors replace “researchers” with the (cited) authors’ last names? 

25. In Table 1, the authors indicate that n_strip is the same as n_eff. Is this true? 

26. In line 183, the word “process” should be changed to “processed”. 

27. The expression “combined with Figure 7” is ambiguous. Please rephrase it for clarity. 

28. Table 2 is not explained. Please describe and explain the results in Table 2. 

29. In line 196, “calculated” should be “calculate”. 

30. How does Figure 8 aid understanding except by showing the position of the monitor? 

31. In Figure 8, what quantity does the color represent? 

32. The paragraph in lines 215—217 is hard to understand. Please rephrase it for better readability. 

Author Response

Dear Editor,

Thank you for your mail from July 4th regarding the review results of manuscript id: materials-1808104: "1 × 4 Wavelength Demultiplexer C-Band Using Cascaded Multimode Interference on SiN Buried Waveguide Structure". We have revised the manuscript while addressing all the comments made by the reviewers. We added new results and clarification text. Our detailed reply to the comments made by the reviewers can be seen below. I hope that in its revised form you may find the manuscript suitable for publication in Materials.

Reviewer 2:

The authors designed a C-band demultiplexer using a cascade of three MMIs based on computational means. They adopted a SiN waveguide buried in SiO2 to reduce the back reflection loss, and optimized the dimensions of the demultiplexer carefully to yield competitive characteristics. This work is a unique contribution to the field. However, the quality of the presentation is far below the standard of publication. For this reason, I do not recommend its publication in the journal Materials. I have the following comments for the authors to consider to improve the manuscript. 

Major comments

  1. The authors wrote that their design is a novel design. Could the authors explain the novelty of their design in more explicit manner?

Answer:

We thank the reviewer for his comment. Although the use of silicon nitride (SiN) material nor MMI technology is not new, the novelty is the combination of the use of SiN as the core material for the device structure and MMI technology to reduce the back reflection loss. Thus, the novelty is the study of the light coupling mechanism using SiN MMI coupler waveguides to function as a four-channel demultiplexer over the C-band. To the best of our knowledge, there are no other articles dealing with this subject.

Reviewer 2:

  1. In line 127, the authors mentioned BPM and FDTD tools of Rsoft photonic CAD software. Can the authors provide necessary simulation parameters for reproducing the results?

Answer:

We thank the reviewer for his comment. According to the reviewer's suggestion, we added the necessary simulation parameters for reproducing the results.

Reviewer 2:

  1. In line 209, the authors wrote that "our design has the best back reflection loss". The description of "best" is misleading since 1) The back reflection loss of the design in this work is higher than that in Ref 28, and 2) There are only two values (not many) of back reflection loss available for comparison.

Answer:

We thank the reviewer for his comment. The issue was fixed according to the reviewer's suggestion.

Minor comments

Reviewer 2:

  1. In line 15, the verb “is” is missing between the words “which” and “usually”.

  1. The sentence in line 31 is hard to understand. The authors might want to change the verb “proposed”.

  1. In line 43, the expression “reflections can be grouped into two mechanisms” is problematic. After grouping, reflections are still reflections but not mechanisms.

  1. In line 49, “Silicon-Nitride” should be “silicon nitride” where the hyphen is symbol removed, and the initial letters are lower case.

  1. In line 50, the expression “… using a SiN [19] buried waveguide 50 based MMI coupler structures for …” is confusing. Do you mean “… using SiN-buried-waveguide-based MMI coupler structures for …”? Note that the article “a” is removed.

  1. In line 55, the initial letter of “Insertion” should be lower case.

  1. It is better to use active voice in the sentence “In this paper, … is presented.” in lines 59—61.

  1. In line 64, should “S-bands” be “S-bends”?

  1. In lines 81 and 83, the phrase “between … to …” should be either “between … and …” or “from … to …”.

  1. In Fig. 1a, the colored label is missing.

  1. In line 96, “is producing” can be just “produces”.

  1. In line 97, it is hard to understand what “the point of two lowest-order modes” is.

  1. In Eq. (1), the superscript is missing for L_\pi.

  1. In lines 99—100, the expression “the effective refractive index of the electrical fundamental mode in the SiN area” can be rewritten as “the effective refractive index of SiN for the electrical fundamental mode”.

  1. In line 105, “MMI” should be “MMI1 and MMI2”.

  1. In lines 107, 115, and 117, “Where” should be “where” in which the initial letter is lower case.

  1. In lines 105—109, the two paragraphs can be a single paragraph.

  1. In line 110, one can replace “There is an option” with “It is possible”.

  1. In line 114, “losses” should be “loss”.

  1. In line 120, can the authors replace “researchers” with the (cited) authors’ last names?

  1. In Table 1, the authors indicate that n_strip is the same as n_eff. Is this true?

  1. In line 183, the word “process” should be changed to “processed”.

  1. The expression “combined with Figure 7” is ambiguous. Please rephrase it for clarity.(line 184-185)

  1. In line 196, “calculated” should be “calculate”.

  1. The paragraph in lines 215—217 is hard to understand. Please rephrase it for better readability.

Answer:

We thank the reviewer for his comments. All of the issues were fixed. Furthermore, a thorough check was performed on grammar, spelling, and punctuation for the whole manuscript.

Reviewer 2:

  1. In Fig. 1b, adding tics for both X and Z axes help readers to understand device dimensions.

Answer:

We thank the reviewer for his comment. This is only a schematic figure for illustration purposes. In Fig. 6 (a-d) there are tics in both X and Z.

Reviewer 2:

  1. Table 2 is not explained. Please describe and explain the results in Table 2.

Answer:

We thank the reviewer for his comment. The results in Table 2. have been described as the reviewer suggested.

Reviewer 2:

  1. How does Figure 8 aid understanding except by showing the position of the monitor?

Answer:

We thank the reviewer for his comment. Figure 8 illustrates the simulation methodology that was used in order to measure back reflection. It is important to understand that the measurement was done in the opposite direction. Moreover, the convergence time was doubled because we wanted to let the light reflect back into the monitor.

Reviewer 2:

  1. In Figure 8, what quantity does the color represent?

Answer:

We thank the reviewer for his comment. The color represents the normalized power intensity in the FDTD simulation. This is the default representation of R-Soft simulation software.

Round 2

Reviewer 1 Report

Thank you for complying with all comments and suggestions.

Reviewer 2 Report

The authors have addressed my comments properly. I recommend the current work for publication in Materials.